# Effects of Maca on Muscle Hypertrophy in C2C12 Skeletal Muscle Cells

**DOI:** 10.3390/ijms23126825

**Published:** 2022-06-19

**Authors:** Dong Yi, Maki Yoshikawa, Takeshi Sugimoto, Keigo Tomoo, Yoko Okada, Takeshi Hashimoto

**Affiliations:** Faculty of Sport and Health Sciences, Ritsumeikan University, 1-1-1 Nojihigashi, Kusatsu 525-8577, Japan; anotherd2018@gmail.com (D.Y.); milktea.maki@gmail.com (M.Y.); hadano4249@gmail.com (T.S.); ktomoo@purdue.edu (K.T.); okd45okd25@gmail.com (Y.O.)

**Keywords:** maca, sarcopenia, muscle hypertrophy, aging, protein

## Abstract

With aging, sarcopenia and the associated locomotor disorders, have become serious problems. The roots of maca contain active ingredients (triterpenes) that have a preventive effect on sarcopenia. However, the effect of maca on muscle hypertrophy has not yet been investigated. The aim of this study was to examine the effects and mechanism of maca on muscle hypertrophy by adding different concentrations of yellow maca (0.1 mg/mL and 0.2 mg/mL) to C2C12 skeletal muscle cell culture. Two days after differentiation, maca was added for two days of incubation. The muscle diameter, area, differentiation index, and multinucleation, were assessed by immunostaining, and the expression levels of the proteins related to muscle protein synthesis/degradation were examined by Western blotting. Compared with the control group, the muscle diameter and area of the myotubes in the maca groups were significantly increased, and the cell differentiation index and multinucleation were significantly higher in the maca groups. Phosphorylation of Akt and mTOR was elevated in the maca groups. Maca also promoted the phosphorylation of AMPK. These results suggest that maca may promote muscle hypertrophy, differentiation, and maturation, potentially via the muscle hypertrophic signaling pathways such as Akt and mTOR, while exploring other pathways are needed.

## 1. Introduction

Sarcopenia is a progressive and age-related decline in skeletal muscle mass and is associated with falls, physical disability, and mortality [1]. Healthy adults lose approximately 8% of their muscle mass every 10 years after the age of 40, and this rate will accelerate to 15% every 10 years after the age of 70 [2]. From a historical point of view, the aging population is a new problem, but projections indicate that by 2050, there will be more people aged 60 or older than adolescents aged 10–24 [3]. The muscles of elderly people have major defects in their regulatory and maintenance abilities during feeding and exercise [4]. In particular, age-related anabolic resistance may show up after prandial in the elderly [5], suggesting the importance of effective nutritional strategies against muscle wasting. Therefore, it is valuable to find a way to enhance the effects of exercise through nutrition or by simply relying on the nutrient intake to slow sarcopenia in elderly people.

*Lepidium meyenii Walp* (maca) grows at altitudes of 2800–5000 m above sea level and was probably domesticated between the years 4000–1200 BCE in Peru [6]. In human studies, it is already known that maca improves menopausal symptoms [7]. In other adult human subjects, maca improves mood, energy, and health status and reduces the chronic mountain sickness score (CMS), a kind of pathology that is observed in only people living at high altitudes [8]. In addition, as a typical lipid-soluble ingredient for maca, macamide extends the weight-loaded swimming time to exhaustion and improves the swimming endurance capacity by reducing the amounts of lactate, lactate dehydrogenase and blood ammonia, such that its antifatigue property has been proven [9]. Furthermore, in the forced swimming test, reduced immobility time indicated that maca has antidepressant activity [10].

*Triterpenoid saponins* have been suggested to have a muscle hypertrophy effect, as they may contribute to maintaining skeletal muscle mass and decreasing diet-induced obesity and muscle wasting [11,12,13]. Ursolic acid is a triterpenoid and is considered a treatment for skeletal muscle disorders [14]. Previous research has shown that triterpenoid saponins are found in maca [15], such as panaxytriol which enhances muscle protein synthesis [11]. In addition to triterpenoid saponins, maca contains abundant nutrients and various amino acids, such as leucine and arginine [16], which are also related to an increase in muscle mass [17,18,19]. Therefore, we hypothesized that maca may have a muscle hypertrophy-promoting effect.

The aim of this study was to clarify the effect of maca extract on the growth of skeletal muscle cells and its underlying mechanisms.

## 2. Results

### 2.1. Adding Maca Promoted Skeletal Muscle Hypertrophy, Differentiation, and Maturation

After adding maca to the C2C12 skeletal muscle cell culture for 2 days, we found that the maca groups had a better growth trend (Figure 1A). The myotube diameters in the 0.1 and 0.2 maca groups were significantly greater than those in the control (con) group (*p* < 0.01) (Figure 1B). We quantified the myotube areas to ascertain the effect of maca on the C2C12 skeletal muscle morphology, and the myotube areas in the 0.1 and 0.2 maca groups were significantly greater than that in the con group (*p* < 0.01) (Figure 1C). The differentiation index was significantly higher in the 0.1 and 0.2 maca groups than in the con group (*p* < 0.01 and *p* < 0.05, respectively) (Figure 1D). The multinucleation was significantly higher in the 0.1 and 0.2 maca groups than in the con group (*p* < 0.01) (Figure 1E).

### 2.2. Effect of Maca on Protein Expression Associated with Muscle Hypertrophy, Differentiation, and Maturation

There was a significant difference in the expression of MyHC-fast between the con group and the 0.2 maca group but there was no significant difference between the con group and 0.1maca group (Figure 2A). There was no significant difference in the expression of MyHC-slow between the con group and the maca groups (Figure 2B). No significant difference was found regarding the expression of MyoD and myogenin, which are related to muscle cell differentiation (Figure 2C,D). The expression of JunB which promotes muscle hypertrophy was not significantly different. The expression of myostatin, MAFbx, and MuRF-1 associated with muscle atrophy was also not significantly different.

### 2.3. Effect of Maca on Protein Expression Related to Muscle Protein Synthesis and Metabolism

The phosphorylation of Akt in the 0.2 maca group tended to be higher than that in the con group (*p* = 0.07) (Figure 3A). The phosphorylation of mTOR in the 0.2 maca group was significantly higher than that in the con group (*p* < 0.05) (Figure 3B). There was no significant difference between the con group and the maca group regarding the phosphorylation of p70S6K (Figure 3C). There was no significant difference between the con group and the maca group in the phosphorylation of 4E-BP-1 (Figure 3D). The phosphorylation of AMPK in the 0.1 maca group (*p* < 0.05) and 0.2 maca group (*p* < 0.01) was significantly higher than that in the con group (Figure 3E).

## 3. Discussion

In this study, we hypothesized that the addition of maca would enhance muscle synthesis, prevent muscle degradation and hence lead to muscle hypertrophy. As a result, after the addition of maca to myotubes for 2 days, the immunohistochemical results showed that adding maca to C2C12 skeletal muscle cell culture enhanced the muscle diameter, myotube area, differentiation index, and multinucleation, suggesting that maca has the potential to promote muscle cell hypertrophy, differentiation, and maturation.

Previous studies have shown that ursolic acid, a kind of triterpene, combined with leucine potentiates the differentiation of C2C12 skeletal muscle cells through the mTOR pathway [20]. Plant-derived ursolic acid can also prevent muscle wasting stimulated by excessive dexamethasone in C2C12 skeletal muscle cell culture [13]. In animal studies, ursolic acid and low-intensity treadmill exercise significantly decreased the expression of MuRF-1 and atrogin-1, thereby attenuating muscle atrophy [21]. Similarly, maslinic acid, pentacyclic triterpene found in olives, also attenuated denervation-induced muscle atrophy via the suppression of MuRF-1 and atrogin-1 [22]. Furthermore, maslinic acid promoted muscle hypertrophy via mTOR signaling [23]. Since maca also contains triterpenes, we predicted that the addition of maca would increase the mTOR pathway, such as the phosphorylation of Akt, mTOR, and p70S6K, in C2C12 skeletal muscle cell culture and lead to muscle hypertrophy. We found that the phosphorylation of Akt in the 0.2 maca group tended to increase, and the phosphorylation of mTOR in the 0.2 maca group was significantly higher than that in the con group, although the Akt-mTOR pathway’s downstream target, p70S6K (Figure 3C) and 4E-BP-1 (Figure 3C) [24] were not changed. Then, we further explored the possibility that there might be a pathway independent of Akt-mTOR that promoted muscle hypertrophy, such as JunB [25]. As a result, there was no significant difference in JunB expression (Figure 2E) and owing to no further research regarding the connection between JunB and muscle hypertrophy, we will not elaborate on it here. Of course, there may also be some other signaling pathways that influence muscle hypertrophy to maca treatment, such as decreased myostatin expression, as shown in ursolic acid treatment [13]. Unfortunately, there was no significant difference in myostatin expression (Figure 2F). In this regard, further studies such as omics (e.g., transcriptome and/or proteome) studies are needed to explore underlying mechanisms for maca treatment.

Interestingly, in this study, the addition of maca regulated skeletal muscle fiber type, especially MyHC-fast (Figure 2A). Previous studies showed that quercetin significantly increased the protein expression of MyHC-slow and significantly decreased MyHC-fast protein expression [26]. Since quercetin is also found in maca [27], we considered that maca may also have affected MyHC-slow, but no significant difference was detected in this study. Overexpression of the muscle-specific E3 ubiquitin-ligase enzymes MuRF-1 and MAFbx lead to skeletal muscle atrophy [28]. Since several previous animal studies have shown that ursolic acid, a kind of triterpene, inhibits muscle mass atrophy [12,29,30,31], we considered that maca also has an inhibitory effect on muscle atrophy, but the results here so no effect on the expression of MuRF-1 and MAFbx (Figure 2E,F). In summary, we suggest that the addition of maca promotes muscle synthesis at least partly via upregulating the Akt-mTOR pathway but the effect on the inhibition of muscle atrophy needs to be verified in future studies.

AMPK greatly affects skeletal muscle development and growth [32]. In this study, we found that the two different concentrations of maca promoted an increase in AMPK phosphorylation in skeletal muscle cell culture (Figure 3D). Basically, activated AMPK inhibits protein synthesis in skeletal muscle by downregulating the mTOR pathway [33] and negatively regulates myotube hypertrophy [34]. On the other hand, our recent studies showed that AMPK phosphorylation is accompanied by muscle hypertrophy in moderate hypoxia [35] and promotes attenuation of myotube atrophy by adding fucoxanthinol [36]. As repeated muscle contractions promote the activation of AMPK [37], activated AMPK stimulates glucose uptake and increases fatty acid oxidation [38]. Therefore, we suggest that the addition of maca promoted AMPK phosphorylation, which in turn stimulated cell metabolism without interfering with the muscle hypertrophic effect of maca.

The results of this study showed that the addition of maca promotes skeletal muscle hypertrophy and differentiation. Studies thus far have shown that maca promotes reproductive health, neuroprotection, antioxidants, antifatigue, antitumoral, liver protection, anti-memory impairment, and immune regulation [39,40,41,42,43,44,45,46]. The results of our study corroborate multiple antiaging effects of the natural food maca, which may be helpful to prevent sarcopenia and improve the quality of life of aging people. Of course, in future studies, more in vivo and in vitro studies are needed to demonstrate maca’s functions, such as testing maca in human muscle cell cultures or animal studies. Furthermore, we may need to examine the direct effects of maca extracts (e.g., triterpenoid saponins, amino acid) on skeletal muscle cell hypertrophy, differentiation, and maturation for further studies.

## 4. Materials and Methods

### 4.1. Cell Culture

C2C12 myoblasts (American Type Culture Collection, Manassas, VA, USA; ECACC, Salisbury, UK) were seeded in a 12-well plate at a density of 1 × 10⁵ cells per culture dish and grown in Dulbecco’s modified Eagle’s medium (DMEM; Thermo Fisher Scientific, Waltham, MA, USA) supplemented with 10% fetal bovine serum (FBS; Thermo Fisher Scientific) and 1% antibiotic antimycotic solution (Sigma–Aldrich, St. Louis, MO, USA). When the cells approached 80%–100% confluence, the medium was changed to DMEM with 2% horse serum (Thermo Fisher Scientific), and then the myoblasts were differentiated into myotubes. The medium was changed every two days, and all experimental cultures were carried out in a 37 °C humid air incubator containing 5% CO_2_.

### 4.2. Experimental Design

The experimental design is summarized in Figure 4A. On day 2 of the induction of differentiation, the myotubes were divided into three groups. The control (con) group was incubated normally, and the maca groups were treated with two different concentrations of maca: the “0.1 maca” group was treated with 0.1 mg/mL of maca and the “0.2 maca” group was treated with 0.2 mg/mL of maca for 48 h. On day 4, the three groups were used for biochemical analyses. The maca extract (lot. no. MCP-001) was provided by Bizen Chemical Co., Ltd. (Okayama, Japan). The extracts were obtained by ethanol extraction and spray drying. The maca extract, containing triterpenoid saponins [15], was dissolved in dimethyl sulfoxide (DMSO; Wako, Japan) and the solution mixed with a differentiation medium was passed through a syringe filter (Sartorius, Germany); DMSO without maca was added to the con group.

### 4.3. Immunohistochemistry

All groups of C2C12 myotubes were cultured on coverslips and fixed with phosphate-buffered saline (PBS) containing 3.7% paraformaldehyde for 15 min at room temperature. Fixed myotubes were permeabilized in 0.2% Triton X-100/PBS for 10 min and blocked with 1% albumin from bovine serum, Fraction V, pH 7.0 (BSA) (FUJIFILM Wako Pure Chemical Corporation, Osaka, Osaka, Japan)/PBS. The cells were incubated with a primary antibody against myosin heavy chain (MyHC; 1:250, Sigma–Aldrich; M4276) overnight, washed with PBS, and incubated for 1 h with a secondary antibody conjugated with Alexa 647 (1:100, Molecular Probes, Grand Island, NY, USA; A31571) at 37 °C [47]. The differentiation index (%) was defined as the MyHC-positive nuclei divided by the total nuclei, and DAPI (Thermo Fisher Scientific, Waltham, MA, USA) was used for nucleic acid staining [35,47]. The multinucleation was defined as the nuclei (at least 2) on a myotube being counted and the percentage of myotubes with multi nuclei in each area was calculated [48]. The myotubes on the coverslips were observed and photographed (3 random areas) under a ×20 objective by using a Biorevo BZ-9000 fluorescence microscope (Keyence, Japan) [35,36,47], in which at least 200 myotubes per area, with a total of over 2400 myotubes were measured. The images of the myotubes were analyzed by BZ-II image analysis software (Keyence, Japan). In addition, we performed three independent experiments (n = 3–4 in each group), and the quantitative measurements were used as reported in previous studies. Briefly, in the randomly selected microscope fields, the diameters of at least 200 myotubes in each area were measured at randomly selected 3 locations (e.g., ①, ②, and ③) taken along the length of the myotubes (Figure 4B) [49]. Then, the average diameter of a myotube was considered as the mean of 3 measurements. The enclosed area in the picture (Figure 4C) was considered the area of myotubes [35,36].

### 4.4. Western Blot Analysis

The cells were rinsed with PBS and lysed in RIPA buffer containing 10 mM Tris–HCl pH 7.4, 1% Na deoxycholate, 1% Triton, 150 mM NaCl, 5 mM EDTA, and 0.1% SDS supplemented with phosphatase inhibitor cocktail (Nacalai Tesque, Japan), phenylmethanesulfonylfluoride solution, protease inhibitor cocktail and PhosSTOP phosphatase inhibitor cocktail (Sigma–Aldrich, St. Louis, MO, USA). Samples were incubated on ice for 1 h and centrifuged at 15,000× *g* for 15 min at 4 °C. To confirm the concentration, the supernatant was collected and analyzed with a Wako protein assay kit (FUJIFILM Wako Pure Chemical Corporation, Japan). Equal amounts of proteins were separated on 8% SDS-polyacrylamide gels at 40 mA for 1 h and transferred to polyvinylidene fluoride membranes at 60 V for 2 h. Blocking One P and Blocking One (Nacalai Tesque, Japan) were used for blocking at room temperature for 30 min. After blocking, the membranes were incubated with primary antibodies against MyHC-fast (Sigma–Aldrich; 1:1000; M4276), MyHC-slow (Abcam, Cambridge, MA, USA; 1:1000; ab11083), myogenin (Santa Cruz; 1:200; sc-12732), MyoD (Santa Cruz; 1:200; sc-32758), TRIM63 (MuRF-1) (Santa Cruz; 1:100; sc-398608), MAFbx (Santa Cruz; 1:100; sc-166806), phosphorylated-mTOR (Ser2448, Cell Signaling Technology; 1:1000; #5536), mTOR (Cell Signaling Technology; 1:1000; #2983), phosphorylated-Akt (Ser473, Cell Signaling Technology; 1:1000; #9271), Akt (Cell Signaling Technology, Danvers, MA, USA; 1:1000; #9272), phosphorylated-p70S6K (Thr389, Cell Signaling Technology; 1:1000; #9205), p70S6K (Cell Signaling Technology; 1:1000; #2708), phosphorylated-4E-BP-1 (Thr37/46, Cell Signaling Technology; 1:1000; #9459), 4E-BP-1 (Cell Signaling Technology; 1:1000; #9644), phosphorylated-AMPK (Thr172, Cell Signaling Technology; 1:1000; #2535), AMPK (Cell Signaling Technology; 1:1000; #2793), myostatin (Santa Cruz; 1:200; sc-34781), JunB (Abcam; 1:200; ab128878), and GAPDH (Sigma–Aldrich; 1:10,000; G9545). Immunoreactive proteins were incubated with anti-mouse IgG (1:10,000) and anti-rabbit IgG (1:10,000) to detect primary antibody binding. The membranes were washed three times with TBST for 10 min each time and treated with the Luminata Forte Western HRP substrate (Millipore) to visualize the bands under the FUSION FX7 EDGE (Vilber Lourmat, Collégien, France), and ImageJ software was used for quantification of the band intensities [50]. A minimum of three independent experiments (n = 4) were performed. As supplementary experiments, we used an independent experiment (n = 3) to detect the protein expression of myostatin and 4E-BP-1.

### 4.5. Statistical Analysis

The data are presented as the means ± S.D. Data were assessed by one-way analysis of variance (ANOVA) with Tukey’s or Games-Howell tests. A value of *p* < 0.05 was considered statistically significant.

## 5. Conclusions

The addition of maca promoted muscle hypertrophy, differentiation, and maturation in skeletal muscle cell culture. Adding maca to skeletal muscle cell culture regulated the expression of MyHC and promoted the phosphorylation of muscle hypertrophic proteins such as Akt and mTOR and the metabolic signal AMPK. Based on the results of immunohistochemistry and increased phosphorylation of Akt and mTOR, we suggest that the addition of maca appears to be effective in the treatment of sarcopenia. Yet, the limitation of the study is that it did not find sufficient molecular mechanisms to explain the effect of maca on skeletal muscle cell hypertrophy, differentiation, and maturation, so we suggest that further research is needed to explore other signaling pathways that influence muscle hypertrophy, differentiation, and maturation to maca treatment by means of omics studies.

## Figures and Tables

**Figure 1 ijms-23-06825-f001:**
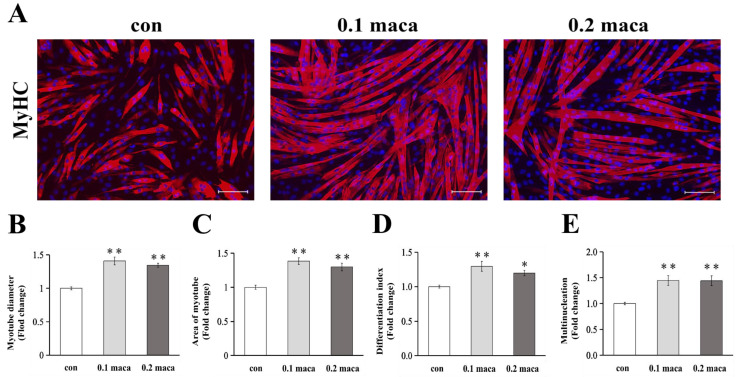
C2C12 myotubes were immunostained with myosin heavy chain (MyHC; red), and the nuclei were immunostained with DAPI (blue). Scale bar: 100 μm. Representative images were taken (**A**). Two independent experiments (n = 7) showed the same trend in myotube diameter (**B**), myotube area (**C**), differentiation index (**D**), and Multinucleation (**E**). * *p* < 0.05 vs. con; ** *p* < 0.01 vs. con.

**Figure 2 ijms-23-06825-f002:**
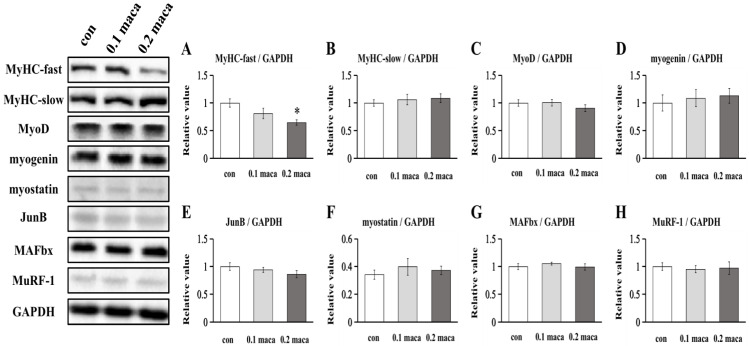
Expression of proteins in C2C12 myotubes. Representative immunoblots are shown on the left side, and the quantification of MyHC-fast (**A**), MyHC-slow (**B**), myogenin (**C**), MyoD (**D**), JunB (**E**), myostatin (**F**), MAFbx (**G**), and MuRF-1 (**H**) was determined after normalizing to GAPDH. * *p* < 0.05 vs. con.

**Figure 3 ijms-23-06825-f003:**
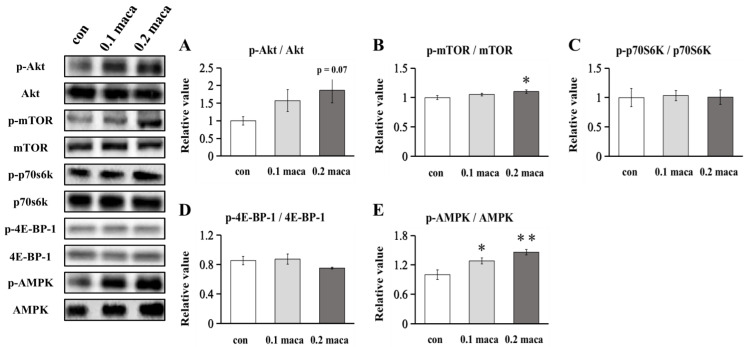
Protein expression in C2C12 myotubes. Representative immunoblots are shown on the left side, and the quantification of Akt (**A**), mTOR (**B**), p70S6K (**C**), 4E-BP-1 (**D**) and AMPK (**E**) phosphorylation was compared between the con group and maca groups. * *p* < 0.05 vs. con; ** *p* < 0.01 vs. con.

**Figure 4 ijms-23-06825-f004:**
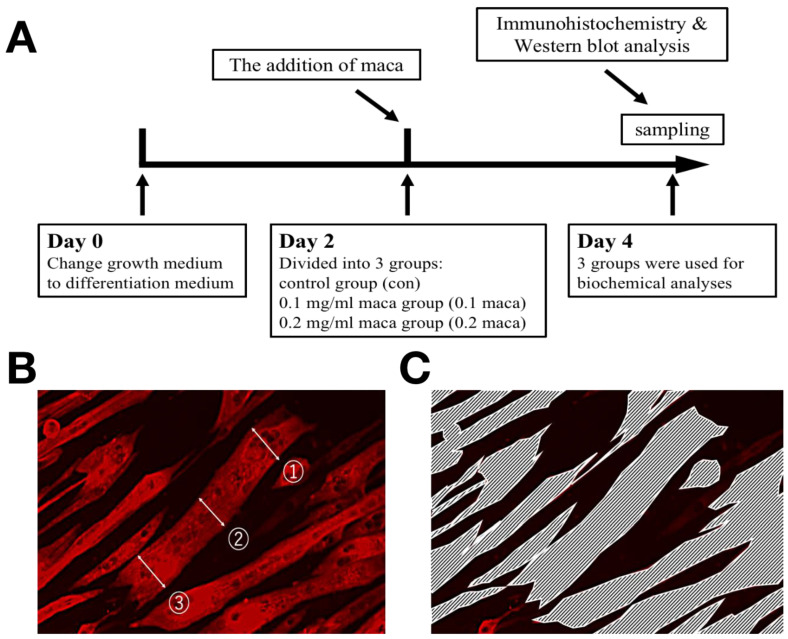
Experimental scheme (**A**) and methods for measuring myotube cell diameter (**B**) and area (**C**) in the experiments.

## Data Availability

Not applicable.

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
