# Peer review of "Effects of Maca on Muscle Hypertrophy in C2C12 Skeletal Muscle Cells"

_ijms, 2022, doi:10.3390/ijms23126825_

Round 1
Reviewer 1 Report
The revised version is satisfactory and I have only one comment concerning the experiment shown in Fig. 4: translation efficiency is determined by the incorporation of OPP into newly synthesized proteins, there are commercial kits to measure translation efficiency (Click-iT protein synthesis assay kit). The method used in Figure 4 is old and inaccurate. Moreover, for muscle cell culture, a bulk protein extract is inaccurate, as differences in translation efficiency are found between fused and non-fused cells (https://doi.org/10.1093/gerona/glac058).
I suggest the authors removing Fig. 4 and the statements related to this Figure (better less and more accurate data).
Author Response
We are so grateful to the reviewer's valuable comments.
Given the comments from the reviewer, we have deleted the data of puromycin expression and related description.
Thank you very much again.
Reviewer 2 Report
The authors followed all of my suggestions and the manuscript in its current form is suitable for publication in this journal.
Author Response
We are so grateful to the reviewer's valuable comments. We have improved our manuscript. Thank you very much again.